# Pain catastrophizing as a mediator of the relationship between pain intensity and depression: Evidence from chronic pain patients in Gaza

## Research Article

chronic pain; pain catastrophizing; depressive symptoms; pain intensity; mediation analysis

**Corresponding author:**
Maha AbuZarifa;
Email: mahazarifamahazarifa@gmail.com

Abdallah AbuJlambo[1], Maha AbuZarifa[2] ⬤, Rasha Alsaadawi[3], Yara Ashour[4], Hanne Lossius[5] and Guido Veronese[6] ⬤

[1]ICU, Al-Shifa Hospital, Palestinian Territory, Occupied; [2]Al-Quds University, Al-Azhar Branch, Palestinian Territory, Occupied; [3]Department of Biostatistics, Virginia Commonwealth University, USA; [4]Al-Quds University, Palestinian Territory, Occupied; [5]Department of anaesthesia and intensive care, Haukeland University Hospital, Norway and [6]Università degli Studi di Milano-Bicocca, Italy

## Abstract

**Background:** Chronic pain represents a major global public health issue. It is associated with wide-ranging psychosocial consequences. Extensive evidence has demonstrated that pain catastrophizing (PC) contributes to the bidirectional association between chronic pain and psychological distress. The present study aims to explore the psychological and cognitive correlates of chronic pain among individuals living in Gaza.

**Methods:** A cross-sectional study was conducted among 272 adults with chronic musculoskeletal pain. Spearman's correlations assessed associations between pain intensity, catastrophizing and depressive symptoms. Multiple regression and bootstrapped mediation analyses (5,000 resamples and PROCESS macro) evaluated predictors and the mediating role of catastrophizing in the pain–depression relationship.

**Results:** Pain intensity was positively correlated with depression ($r = 0.28$, $p < 0.001$) and catastrophizing ($r = 0.39$, $p < 0.001$). A stronger correlation was found between catastrophizing and depression ($r = 0.54$, $p < 0.001$). Mediation analysis demonstrated that catastrophizing fully mediated the association between pain intensity and depression (indirect effect = 0.95, 95% confidence interval = [0.65–1.29]).

**Conclusion:** PC is a key psychological mechanism linking pain intensity and depression among patients with chronic pain in Gaza. Integrating cognitive-behavioral therapy, mindfulness and emotion regulation strategies into pain management may improve mental health outcomes in conflict-affected settings.

## Impact statements

Chronic pain is one of the leading causes of years lived with disability worldwide. In regions experiencing humanitarian crises, chronic pain and related mental health conditions are likely to deteriorate. Pain catastrophizing (PC) is a key factor in the relationship between psychological distress and chronic pain. However, when it comes to managing chronic pain in Gaza, there is a lack of epidemiological data and sufficient clinical guidelines. This study provides evidence that PC is a key psychological mechanism linking pain intensity and depression among patients with chronic pain in Gaza. The findings suggest that PC should be a primary focus of clinical management in patients with chronic pain, particularly in regions affected by conflict. The regular integration of cognitive-behavioral therapy, mindfulness-based practices and emotion control techniques into standard chronic pain management plans is highly supported by this study. Furthermore, the results will have important implications for clinical practice and public health policy in Palestine. By informing the development of integrative, culturally sensitive and evidence-based interventions, the study aims to support health professionals, institutions and policymakers in more effectively responding to the complex needs of individuals experiencing chronic pain in the Gaza Strip.





## Background

Chronic pain represents a major global public health issue and is one of the leading causes of years lived with disability worldwide (Vos et al., 2015). It is associated not only with substantial individual suffering but also with wide-ranging psychosocial consequences, including increased risk of depression, anxiety, reduced quality of life and impaired social and family functioning

(Denkinger et al., 2014; Dueñas et al., 2016; Tsuji et al., 2016; Müller et al., 2017). The co-occurrence of chronic pain and mental health disorders is well-documented, with prevalence estimates ranging from 5% to 90%, depending on the population and methodology used (Velly and Mohit, 2018). For example, rates of comorbidity range between 15% and 30% among Asian American populations (Kim et al., 2015), and between 30% and 50% in other demographic groups (Kroenke et al., 2011). Some studies report rates as high as 60% in particularly vulnerable populations (Doan et al., 2015; Campos et al., 2020), emphasizing the critical need for integrated clinical approaches that simultaneously address physical and psychological dimensions of chronic pain.

In contexts of political violence and prolonged humanitarian crises – such as the Gaza Strip – chronic pain and associated mental health conditions are likely to be exacerbated. However, there is a notable lack of epidemiological data and contextually appropriate clinical guidelines regarding the assessment and treatment of chronic pain in Gaza. This gap underscores the urgency and relevance of the present study.

A large body of research has emphasized the reciprocal relationship between chronic pain and depression (Bair et al., 2003). According to cognitive-behavioral theories, pain itself is not sufficient to trigger depressive symptoms; rather, the individual's cognitive appraisal and coping strategies are key mediating factors in this relationship (Rudy et al., 1988). Maladaptive cognitive-emotional patterns – such as feelings of helplessness, passive coping, avoidance, emotion-focused strategies and low self-efficacy – have been linked to poorer pain adjustment and greater psychological distress (Covic et al., 2003).

One of the most influential cognitive constructs implicated in this process is pain catastrophizing (PC). It refers to a set of maladaptive cognitive and affective responses to pain, characterized by an exaggerated negative orientation toward actual or anticipated pain experiences (Breivik et al., 2006). PC comprises three core dimensions: rumination, persistent and intrusive thoughts about pain; magnification, involving exaggerated perceptions of the threat posed by pain; and helplessness, reflecting a sense of loss of control and inability to manage pain (Sullivan et al., 2001).

Extensive empirical evidence has demonstrated that PC plays a central role in the bidirectional association between chronic pain and psychological distress (Richardson et al., 2009). For instance, among older adults with chronic pain in Australia, higher levels of catastrophizing have been associated with greater perceived pain intensity and more severe depressive symptoms (Wood et al., 2013). Similarly, studies involving Korean-American immigrants have shown that PC mediates the relationship between depressive symptoms and negative pain-related outcomes (Kim et al., 2015). Notably, the helplessness component of catastrophizing has been found to specifically mediate the association between pain severity and the intensity of depressive symptoms (Sánchez-Rodríguez et al., 2020). These findings underscore the relevance of catastrophizing as a transdiagnostic cognitive-emotional risk factor, as well as a promising target for intervention through cognitive-behavioral therapy (CBT) and other integrative psychosocial treatments.

## Theoretical framework

This study draws upon the cognitive-behavioral model of chronic pain, which posits that cognitive appraisals significantly shape emotional and behavioral responses to pain, thereby influencing overall pain experience and psychological adjustment (Linton & Shaw, 2011). In particular, PC is conceptualized within this framework as a dysfunctional cognitive schema that not only amplifies perceived pain but also increases vulnerability to depressive and anxious symptomatology (Sullivan et al., 2001). According to this model, individuals with high levels of catastrophizing are more likely to engage in avoidance behaviors, interpret pain as threatening and unmanageable and experience heightened emotional distress. These processes can create a self-reinforcing cycle of pain, functional disability and psychological suffering, particularly in socio-politically adverse environments (Wood et al., 2013).

In war-affected contexts such as Gaza, where individuals are exposed to chronic stress, instability and trauma, the cognitive appraisal of pain may be further influenced by contextual and existential threats. This may exacerbate catastrophizing tendencies and intensify the emotional burden of chronic pain by reinforcing perceptions of uncontrollability and limited coping resources. This reinforces the catastrophizing, which, in turn, intensifies the depressive symptoms among chronic pain sufferers, emphasizing the importance of the relevance of examining PC as a key psychological variable linking pain intensity to depression in culturally and contextually relevant humanitarian settings.

## Objectives and hypotheses

The present study aims to explore the psychological and cognitive correlates of chronic pain among individuals living in the Gaza Strip, a region marked by protracted conflict, socioeconomic hardship and limited access to healthcare services.

### Specific objectives

1. To estimate the prevalence and severity of depression, anxiety and stress among individuals experiencing chronic pain.
2. To assess levels of PC and its specific dimensions (rumination, magnification and helplessness).
3. To examine the relationships between pain intensity, PC and psychological distress (depression, anxiety and stress).
4. To evaluate the potential mediating role of PC in the relationship between pain intensity and depression.

### Hypotheses

**H1:** Higher levels of pain intensity will be significantly associated with higher levels of depressive, anxious and stress-related symptoms.

**H2:** PC will be positively correlated with both pain intensity and psychological distress.

**H3:** PC will mediate the relationship between pain intensity and depression, such that greater pain intensity will predict higher depression scores indirectly through increased levels of catastrophizing.

## Materials and methodology

This cross-sectional study was conducted among patients with chronic pain syndrome attending the rheumatology, orthopedics and neurology outpatient clinics at Al-Shifa Hospital, Gaza City, between September 3 and 18, 2022. These clinics serve as the primary referral centers for patients with chronic pain in the region.

The minimum required sample size ($n = 107$), using G power analysis software (*F*-test, multiple linear regression, $f^2 = 0.15$, $\alpha = 0.05$,

power = 0.95). Because of the lack of a centralized registry, consecutive random sampling was applied. Of the 318 patients approached, 299 agreed, with a 93.9% response rate. There were 272 patients finally included after applying exclusion criteria.

### Eligibility criteria, data collection and study measures

Adult patients ≥18 years with chronic musculoskeletal pain (>3 months) who were able to communicate in Arabic were included. The exclusion criteria were patients with cancer-related pain, gastrointestinal pain and primary psychiatric disorders such as major depression and psychosis. Autoimmune/systemic inflammatory disease (rheumatoid arthritis and lupus), disc herniation, is the main cause of cognitive impairment.

Data were gathered by trained research assistants who conducted structured face-to-face interviews using validated Arabic versions of standardized scales. Those scales are related to the study variables. A comprehensive statistical approach was undertaken to examine the relationships among the primary study variables: pain intensity, PC and depressive symptoms, as measured by the Depression Anxiety Stress Scale-21 (DASS-21). Given the non-normal distribution of the data, nonparametric and robust analytical methods were applied throughout. Data included pain duration and intensity, Numerical Rating Scale (NRS), Pain Catastrophizing Scale (PCS) and DASS-21 scale for mental health disorders. The Arabic versions of the PCS and DASS-21 have been previously validated in Arabic-speaking clinical populations, including Palestinian cohorts, demonstrating good reliability and construct validity. Established cutoff scores from these validations were applied in the present study. For pain intensity, we used the standard severity categories (mild: 1–3, moderate: 4–6 and severe: 7–10) reported for descriptive purposes, while PC was defined as a PCS score > 30. For the DASS-21, the established cutoff values were applied (≥10 for depression, ≥8 for anxiety and ≥ 15 for stress; Alghadir et al., 2016).

### Ethics

Approval was granted by the Palestinian Ministry of Health Research Ethics Committee (No. 688733). Written informed consent was obtained from all participants in accordance with the Declaration of Helsinki.

### Statistical analysis

#### Correlation analysis

Spearman's rank-order correlation was conducted to explore the associations between subjective pain experiences, PC and depression. This bivariate analysis assessed the strength and direction of monotonic relationships among study scales: NRS, PCS and the Depression subscale of DASS-21. (Spearman, 1904).

Bivariate associations were examined using both correlation coefficients and visual inspection of scatterplots (see Supplementary Figures S1–S5) to ensure the assumptions of linear relationships were met.

### Mediation analysis

To examine whether PC mediates the relationship between pain intensity and depressive symptoms. Although the conceptual framework for mediation was informed by the approach described by Baron and Kenny (1986), the indirect effects in the present study were tested using bootstrapping procedures with bias-corrected

confidence intervals (CIs), consistent with current best practice recommendations for mediation analysis (Hayes, 2018). We employed the causal-steps approach outlined by Baron and Kenny (1986), consistent with previous research in chronic pain populations (Wood et al., 2013). According to this framework, mediation is established when: (1) the independent variable (pain intensity, NRS) significantly predicts the mediator (pain catastrophizing, PCS); (2) the independent variable significantly predicts the dependent variable (depression, DASS-21); (3) the mediator significantly predicts the dependent variable when controlling for the independent variable; and (4) the effect of the independent variable on the dependent variable is reduced when the mediator is included in the model.

We conducted a series of regression analyses to test these conditions. First, we examined the total effect (c path) of pain intensity on depression. Second, we tested whether pain intensity predicted PC (a path). Third, we evaluated the effect of PC on depression while controlling for pain intensity (b path), and the direct effect of pain intensity on depression when controlling for catastrophizing (c' path).

To assess the statistical significance of the indirect effect (a × b), we employed bootstrapping with 5,000 resamples to generate bias-corrected 95% CIs using the 'mediation' package in R (Tingley et al., 2014). This approach does not assume normality of the sampling distribution and provides more accurate CIs than traditional methods (Hayes, 2013). Mediation was considered significant if the CI for the indirect effect did not include zero. Additionally, we calculated the Sobel test as a supplementary measure of indirect effect significance (Sobel, 1982). All analyses were performed using R version 4.4.0 (R: The R Project for Statistical Computing, n.d.).

## Results

### Correlation analysis

There was a statistically significant weak positive correlation between NRS and DASS-21 depression ($r = 0.28$, $p < 0.001$). A moderate positive correlation was found between NRS and PCR ($r = 0.39$, $p < 0.001$). Furthermore, a strong positive correlation was observed between PCS and depression-DASS-21 ($r = 0.54$, $p < 0.001$), suggesting the potential for mediation (Table 1).

Visual inspection of the scatterplots confirmed the linear nature of these relationships (Supplementary Figures S1–S2). The correlation between pain intensity and PC showed a clear positive trend with moderate dispersion (Supplementary Figure S1), while correlations with psychological distress measures showed greater variability (Supplementary Figure S2). Item-level analysis of the DASS-21 revealed that specific items related to stress (Item 1) and difficulty relaxing (Item 12) showed the strongest correlations with pain

**Table 1.** Spearman's correlation matrix among key study variables

| | Pain intensity | Pain catastrophizing | DASS-21 depression |
|---|---|---|---|
| Pain intensity | 1 | 0.39** | 0.28** |
| Pain catastrophizing | | 1 | 0.54** |
| DASS–21 depression | | | 1 |

** p-value < 0.01.

intensity, while items assessing panic (Item 15) and trembling (Item 7) showed weaker associations (Supplementary Figures S3–S5).

### Mediation analysis

The total effect of pain intensity on depression was significant, $B = 1.28$, SE $= 0.267$, $t$ (270) $= 4.831$, $p < 0.001$. Pain intensity significantly predicted the mediator, PC ($B = 2.34$, SE $= 0.313$, $t = 7.48$, $p < 0.001$). When both pain intensity and PC were entered into the model, only PC remained a significant predictor of depression ($B = 0.404$, SE $= 0.046$, $t = 8.858$, $p < 0.001$), while pain intensity became nonsignificant ($B = 0.340$, SE $= 0.258$, $t = 1.318$, $p = 0.189$), indicating full mediation. Bootstrapping results confirmed this indirect effect, with an effect size of 0.9481 and a 95% CI $= [0.6382, 1.2972]$, not crossing zero, affirming the statistical significance.

Table 2 presents the complete mediation analysis results. The analysis revealed that PC fully mediated the relationship between pain intensity and depressive symptoms. The total effect of pain intensity on depression was significant (c path: $B = 1.29$, SE $= 0.27$, 95% CI $= [0.76, 1.81]$, $p < 0.001$), indicating that each unit increase in pain intensity was associated with a 1.29-point increase in depression scores. Pain intensity significantly predicted PC (a path: $B = 2.34$, SE $= 0.31$, 95% CI $= [1.73, 2.96]$, $p < 0.001$), demonstrating that higher pain intensity was associated with greater catastrophizing. When both pain intensity and PC were entered simultaneously as predictors of depression, PC remained a significant predictor (b path: $B = 0.40$, SE $= 0.05$, 95% CI $= [0.31, 0.49]$, $p < 0.001$), while the direct effect of pain intensity became nonsignificant (c' path: $B = 0.34$, SE $= 0.26$, 95% CI $= [-0.17, 0.85]$, $p = 0.189$). This pattern indicates full mediation, as the previously significant relationship between pain intensity and depression was completely accounted for by PC. The indirect effect of pain intensity on depression through PC was 0.95 (SE $= 0.17$, 95% CI $= [0.65, 1.29]$, $p < 0.001$). The CI, derived from 5,000 bootstrap resamples, did not include zero, confirming the statistical significance of the mediation effect. The Sobel test further supported this finding ($z = 5.72$, $p < 0.001$). The proportion of the total effect mediated was 73.6%, indicating that nearly three-quarters of the relationship between pain intensity and depression operated through PC. Figure 1 illustrates the mediation model, showing both the total effect (**Panel A**) and the mediated model (**Panel B**).

The mediation model explained 28.7% of the variance in depression scores (Adjusted $R^2 = 0.282$, $F (2, 269) = 54.25$, $p < 0.001$), with PC emerging as the primary predictor. These findings suggest that the cognitive-emotional appraisal of pain, rather than pain intensity per se, drives the development of depressive symptoms in this population.

**Table 2.** Mediation analysis results: PC as a mediator of the relationship between pain intensity and depression.

| Path | Coefficient | SE | 95% CI | *p*-value |
|---|---|---|---|---|
| Total effect (c) | 1.289 | 0.267 | (0.763, 1.814) | <.0001* |
| Direct effect (c') | 0.340 | 0.258 | (−0.168, 0.849) | 0.1887 |
| Indirect effect (a × b) | 0.948 | 0.166 | (0.652, 1.288) | <.0001* |
| Path a (NRS → PCS) | 2.344 | 0.313 | (1.727, 2.961) | <.0001* |
| Path b (PCS → Depression) | 0.404 | 0.046 | (0.315, 0.494) | <.0001* |

* *p*-value < 0.05.

### Discussion

Our findings support previous research suggesting a strong correlation between the intensity of pain and depressive symptoms for patients with chronic pain. A cross-sectional questionnaire survey indicated that patients with high levels of depression had significantly high scores on the pain scale (Rapti et al., 2019). Our findings are consistent with these data, suggesting a statistically positive correlation between intensity of pain and depressive symptoms, supporting the idea that pain is not a mere sensorial aspect but something highly integrated with psychological processing. These associations provide additional empirical support for the biopsychosocial theory of chronic pain in that it asserts that biological, cognitive and affective factors intertwine in determining individuals' perceptions about pain and disability (Rudy et al., 1988).

Comorbidity between depression and chronic pain was detailed extensively in the literature, and our findings add a cross-cultural dimension by depicting overlapping pathways. Both conditions engage common neurobiological pathways, such as dysfunctions in overlapping networks of the brain and neurotransmitters involved in neuroplasticity (Sheng et al., 2017). Functional neuroimaging revealed shared activity in brain centers, such as the amygdala, nucleus accumbens, prefrontal cortex and anterior cingulate cortex, engaged in both processing mood and processing pain (Ong et al., 2019). Recent neurophysiological studies revealed specific pathways depicting the integrated nature of depressive and chronic pain circuits (Bonilla-Jaime et al., 2022).

In line with the Fear-Avoidance Model of Chronic Pain, we observed a moderate positive correlation between pain severity and PC. This outcome is consistent with a comprehensive meta-analysis, which emphasized that PC is associated with pain intensity; the relationship is not necessarily strong, thus pointing to the greater relevance of functional and mental health outcomes in shaping the pain experience (Crombez et al., 2012). Our findings also revealed a robust association between PC and depressive symptoms, suggesting that maladaptive cognitive responses to pain may contribute more directly to emotional distress than to the sensory component of pain itself (Kim et al., 2015). These results support the hypothesis that PC serves as a cognitive vulnerability factor for psychological impairment, acting as a psychological amplifier of distress, rather than a mere byproduct of pain intensity. Importantly, our mediation analysis revealed that PC fully mediated the relationship between pain intensity and depressive symptoms. This finding aligns with prior research conducted among Asian immigrant populations, which similarly identified a strong mediating role of catastrophizing in the pain–depression relationship (Malfliet et al., 2017), as well as a study on a tertiary-level referral pain center in Australia (Wood et al., 2013). Neuroimaging data from previous studies offer additional support by showing that PC is associated with heightened activation in brain regions involved in pain processing, although not always directly linked to depressive symptoms (Kim et al., 2015).

These results point to a possible neurocognitive pathway through which catastrophic thinking heightens the emotional toll of pain, even in the absence of increased nociceptive input. We further observed a stronger correlation between DASS-21 items related to stress and pain intensity scores, a finding consistent with previous studies that have shown a positive correlation between perceived stress and pain perception. This is attributed to the stress subscale's focus on arousal and tension, which are more linked to the physiological and psychological responses to chronic pain

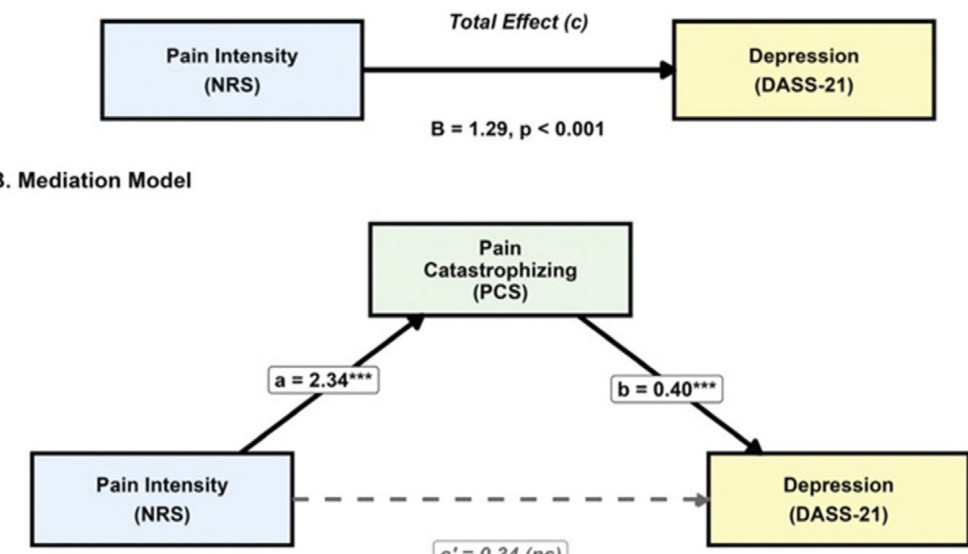

**Figure 1.** The mediation model shows PC as a full mediator of the relationship between pain intensity and depression. **Panel A** shows the total effect model with the direct relationship between pain intensity and depression. Panel B shows the mediation model with PC included, demonstrating that the direct effect becomes nonsignificant (dashed line), while the indirect path through catastrophizing remains significant. Values represent unstandardized regression coefficients. ***$p$ < 0.001; ns = not significant.

(Taylor et al., 2005). However, a weaker correlation emerged between pain intensity scores in terms of anxiety-related items, which reflects findings from other cross-sectional studies that failed to identify a direct relationship between pain severity and anxiety (Hasan et al., 2021). One possible explanation is that our study included a diverse group of patients with various chronic musculoskeletal pain pathologies. The current war situation in Gaza is alarming for increasing depressive symptoms in general. A study by AbuZarifa et al. (2025) indicates that the prevalence of depressive symptoms across the externally displaced population is 97% (AbuZarifa et al., 2025), and Zughbur et al. (2025) showed that 72.7% of the internally displaced population reported moderate to severe depression symptoms (Zughbur et al., 2025). Similarly, due to the large number of war injuries, chronic pain is expected to become an epidemic problem in Gaza (*Humanitarian Situation Update #326 | Gaza Strip - Occupied Palestinian Territory | Relief-Web*, n.d.), (AbuZarifa et al., 2026. So, it is important to advocate for a public health strategy to provide both physical and mental health services, such as PC, depression screening and treatment. Thus, the findings of the present study suggest that addressing PC may be a critical mechanism for improving pain-related and psychological outcomes in the expected large number of wounded people experiencing chronic pain due to the current conflict. From a clinical perspective, the full mediating role of PC highlights the need to integrate targeted psychological interventions focused specifically on maladaptive pain-related cognitions. There was strong emerging evidence that supports the efficacy of emotion-focused cognitive therapies that target both negative affect and dysfunctional emotion regulation in individuals with chronic pain and concurrent depressive symptoms; by enhancing emotional regulation and disrupting cycles of catastrophizing, these interventions can lead to more adaptive responses to persistent pain (Boersma et al., 2019). In addition, mindfulness techniques are defined as cultivating present-moment awareness and nonjudgmental acceptance of pain, which may interrupt catastrophic appraisal processes. Interventions have been associated with significant reductions in PC and emotional reactivity (Conti et al., 2020). Likewise, spiritual well-being, including a sense of purpose, meaning and inner peace, has been shown to buffer the negative effects of PC and improve overall pain management (Shaygan and Shayegan, 2019).

One study conducted among Palestinian nurses showed a weak knowledge of pain management (Salameh, 2018). As a result, it is crucial to advocate for capacitating health workers' pain assessment skills for chronic pain patients through regular and continuous education and professional development programs. Frequent mental health screening programs targeting PC are recommended for patients with chronic pain. Multidimensional interventions that incorporate cognitive, emotional and existential domains may offer more robust protection against the affective burden of chronic pain. Incorporating these elements into existing chronic pain management frameworks may result in improved psychological resilience and enhanced quality of life for individuals with comorbid chronic pain.

## Limitations

Despite its contributions to the understanding of the psychological mechanisms underlying chronic pain and depression, the present study is subject to several limitations that should be acknowledged when interpreting the results. Initially, the fact that a nonprobabilistic convenience sample is selected from a particular socio-political background, namely, people living in Gaza precludes the generalizability of

results to larger or differently cultured groups. The sociocultural and geopolitical determinants specific to this area can affect both pain experience and psychological morbidity; accordingly, the outcomes may not be maximally transportable to other clinical or community contexts. Second, the model did not incorporate several appropriate biobehavioral and environmental predictors of chronic pain, including participants' levels of physical activity, drug compliance with prescribed analgesic or psychotropic medications, sleep quality and impairment related to pain. Third, the research did not evaluate social and relational variables known to have a pivotal role in pain and emotional distress experience and expression. Factors such as family functioning, social support and quality of relationships potentially moderated the psychological impact of chronic pain but were not part of the model. Finally, the cross-sectional design of the study prohibits making causal inferences of the full mediation effect of PC. Despite statistically tested mediation, the lack of longitudinal data does not allow us to ascertain the temporal precedence of pain severity, catastrophizing and depressive symptoms. Longitudinal designs must be employed in order to determine the direction and stability of these relationships in the long term. Such research in the future should seek to overcome these limitations by using longitudinal designs, including a wider range of biopsychosocial predictors, and enrolling more diverse samples across various cultural and geopolitical settings. While our study examined overall PC, future research should explore the differential contributions of its subdimensions to depressive symptoms, as this may provide more nuanced insights for targeted psychological interventions.

## Conclusion

This study provides compelling empirical evidence that PC plays a pivotal role in the psychological and sensory dimensions of chronic pain. In individuals suffering from chronic pain, catastrophizing was not merely associated with depressive symptoms, but it entirely mediated the relationship between pain intensity and depression. This striking result underscores that the cognitive-emotional interpretation of pain may be more detrimental to mental health than the physical sensation of pain itself. The magnitude of this mediation effect highlights PC as a core target for clinical intervention, not a secondary factor.

In light of this, we strongly advocate for the routine incorporation of CBT, mindfulness-based approaches and emotion regulation strategies into standard chronic pain treatment protocols. These interventions, when tailored to reduce catastrophic thinking, have the potential to dramatically alleviate psychological suffering, decrease pain intensity and improve patients' quality of life. The cost of ignoring the cognitive-affective dimensions of pain is too high, both in terms of human suffering and the burden on health systems.

PC is no longer a peripheral concern in chronic pain care. It is a central clinical priority, and any treatment program that fails to address it risks being fundamentally incomplete.

**Open peer review.** To view the open peer review materials for this article, please visit http://doi.org/10.1017/gmh.2026.10151.

**Supplementary material.** The supplementary material for this article can be found at http://doi.org/10.1017/gmh.2026.10151.

**Data availability statement.** The datasets generated and/or analyzed during the current study are not publicly available due to privacy and ethical restrictions, but are available from the corresponding author on reasonable request.

**Acknowledgments.** The authors would like to thank Mohammed Imad, Mohammed Elsheikh Khalil, Mohammed Al-khaldi, Siham Al-Shawamreh, Islam Ayyash and colleagues who helped with the data access and collection.

**Author contribution.** AA conceptualized the study. AA, YA, MA and RA prepared and refined the data for analysis, executed a statistical evaluation, designed graphical representations and contributed to writing the manuscript. HL and GV evaluated the manuscript, provided key insights on the outcome data and assisted with data interpretation. All authors had full access to the study data and took responsibility for its submission.

**Financial support.** This research received no specific grant from any funding agency in the public, commercial or not-for-profit sectors.

**Competing interests.** The authors declare none.

**Declarations.** Ethics, approval and consent to participate: The Palestinian Ministry of Health Research Ethics Committee granted ethical permission with the serial number 688733. Written informed consent was obtained from all participants before enrolment. Participants were provided with detailed information about the study's purpose, procedures, risks and benefits, and were assured of confidentiality and voluntary participation. All methods were performed in accordance with the relevant guidelines and regulations, including the Declaration of Helsinki.

**Consent for publication.** Not applicable, as this manuscript does not contain any person's data in any form (including individual details, images or videos).

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
