## [Reviewer Report]

The study investigates a highly timely issue — the importance of addressing mental health in chronic pain patients within specific war and humanitarian settings. This cross-sectional study highlights the psychological burden associated with chronic pain and identifies pain catastrophizing as a key mechanism that exacerbates depression among individuals living in crisis-affected areas.

The “theoretical framework” in the introduction is lacking references. I would recommend the authors to add few references to support their claims (page 4, lines 1-23)

Since the specific hypotheses also address the dimensions of pain catastrophizing—such as rumination, magnification, and helplessness—it would be helpful to briefly describe these in the introduction to improve the logical flow and clarity of the hypotheses. Additionally, why is it that only the helplessness dimension is expected to demonstrate the strongest mediating effect? Rumination has also been identified as an important factor in depression, so it would be useful to clarify this distinction. (H4, page 4, line55)

The proposed mediation model is theoretically coherent with the study rationale: pain catastrophizing is appropriately conceptualized as a mechanism linking pain intensity to depression. However, this connection could be articulated more explicitly in the background section, particularly regarding why catastrophizing is salient in war or humanitarian contexts.

Under the eligibility criteria page 5 (line 33), it is not clear this exclusion criteria “The exclusion criteria for pediatric patients include cancer-related pain” but the sample was above 18 years of age so there were no pediatric patients.

Please, report the exact cut-offs scores of the scales that were used in relation to the citations page 5 (lines 50-54).

If you cite Baron & Kenny as your conceptual foundation, make it clear that their model provides the framework for mediation, but your testing method (bootstrapping) follows current best practice.

The mediation analysis is clearly presented and statistically appropriate. However, the interpretation should be more cautious. The claim of “full mediation” is not fully justified given the cross-sectional design, which precludes causal inference. I believe this limitation should be addressed.

The mediation analyses appear to have used only the total PCS score and did not examine the hypothesized effects of specific subcomponents of pain catastrophizing (e.g., helplessness, HP4). I recommend reporting these additional analyses, even if non-significant, as they could offer a more nuanced understanding of which cognitive dimensions contribute most strongly to depression. Exploring these psychological subdimensions may also help identify potential targets for psychological intervention.

In addition, it would be informative to present the raw mean levels of both scales (and their subscales) and compare them with the established cut-off scores described in the Methods section. Indicating whether participants’ scores are above or below clinical thresholds—and how they differ from populations not exposed to war conditions—would provide valuable contextual interpretation and could be discussed in light of existing literature.

Table 2 something went off in the table – please check it again.

Figure 1 Mediation Model B – the numbers are not clearly visible.

---

## [Reviewer Report]

The study is important and relevant. Evidence related to chronic pain assessment and management from underrepresented areas and populations are important and often overlooked. Minor change - Orientation of the text is right to left - please correct. The use of AI is evident in some areas - remove the hyphens as this is trademark AI generation.